# Expectancy-based rhythmic entrainment as continuous Bayesian inference

**Jonathan Cannon** [ID] *

Department of Brain and Cognitive Science, Massachusetts Institute of Technology, Cambridge, Massachusetts, United States of America

* jcan@mit.edu

**Data Availability Statement:** All code used to produce all simulation data and figures is available at https://github.com/joncannon/pippet.

**Funding:** The author(s) received no specific funding for this work.

## Abstract

When presented with complex rhythmic auditory stimuli, humans are able to track underlying temporal structure (e.g., a "beat"), both covertly and with their movements. This capacity goes far beyond that of a simple entrained oscillator, drawing on contextual and enculturated timing expectations and adjusting rapidly to perturbations in event timing, phase, and tempo. Previous modeling work has described how entrainment to rhythms may be shaped by event timing expectations, but sheds little light on any underlying computational principles that could unify the phenomenon of expectation-based entrainment with other brain processes. Inspired by the predictive processing framework, we propose that the problem of rhythm tracking is naturally characterized as a problem of continuously estimating an underlying phase and tempo based on precise event times and their correspondence to timing expectations. We present two inference problems formalizing this insight: PIPPET (Phase Inference from Point Process Event Timing) and PATIPPET (Phase and Tempo Inference). Variational solutions to these inference problems resemble previous "Dynamic Attending" models of perceptual entrainment, but introduce new terms representing the dynamics of uncertainty and the influence of expectations in the absence of sensory events. These terms allow us to model multiple characteristics of covert and motor human rhythm tracking not addressed by other models, including sensitivity of error corrections to inter-event interval and perceived tempo changes induced by event omissions. We show that positing these novel influences in human entrainment yields a range of testable behavioral predictions. Guided by recent neurophysiological observations, we attempt to align the phase inference framework with a specific brain implementation. We also explore the potential of this normative framework to guide the interpretation of experimental data and serve as building blocks for even richer predictive processing and active inference models of timing.

## Author summary

In motor and perceptual tasks involving auditory rhythms, humans show a remarkable proficiency for recognizing an underlying "beat" structure and using it to precisely anticipate the timing of auditory events. Models have been built to describe the faculty of perceptual and motor "entrainment," but they have done little to describe this process in a general

**Competing interests:** The authors have declared that no competing interests exist.

language consistent with other perceptual and cognitive processes. Here, we treat entrainment as the formal problem of estimating the phase and tempo underlying a structured auditory rhythm in real time, based on a set of expectations for what phases of the rhythm are likely to be marked by auditory events. When this problem is solved mathematically, the solution reproduces some surprising nuances of human entrainment. It does so by introducing two new elements that have not been modeled before: uncertainty about phase and tempo, and a systematic biasing effect of strong expectations with the power to distort perceived passage of time. This model of entrainment is a plausible description of what may be happening in motor-associated regions of the brain during rhythm listening.

## Introduction

The human brain is remarkably proficient at identifying and exploiting temporal structure in its environment, especially in the auditory domain. This phenomenon is most easily observed in the case of auditory stimuli with underlying periodicity: humans adeptly and often spontaneously synchronize their movements with such auditory rhythms [1], and human brain activity in auditory and motor regions aligns to auditory stimulus periodicity even in the absence of movement [2]. Both of these phenomena are cases of "entrainment" (sensorimotor and neural, respectively), where we define "entrainment" as in [3]: the temporal alignment of a biological or behavioral process with the regularities in an exogenously occurring stimulus.

A simple sinusoidal phase oscillator can entrain to a periodic stimulus; however, it is difficult to discuss the flexible entrainment of human behavior and cognitive processes to variable and sometimes aperiodic patterns such as speech without invoking the cognitive concept of "temporal expectation." Expectations for event timing can be used to achieve a range of behavioral goals. They can help us hone our sensory detection, our sensory discrimination, and our response time for behaviorally important stimuli at the anticipated time [4–6]. In some situations, temporal expectations attenuate neural responses [7], which may help to conserve neural resources. And timing expectations bias our perception of time, allowing us to use prior experience to supplement noisy sensory data as we make temporal judgments [8].

Entrainment in humans involves an interplay of stimulus and temporal expectation [9]. Nowhere is this clearer than in interaction with music, humankind's playground for auditory temporal expectation and entrainment [10]. But the precise nature of this interplay is an open question. The framework of Dynamic Attending Theory characterizes temporal expectancy as pulses of "attentional energy" issued by entrained neural oscillators, and mathematical models based on these ideas describe bidirectional interactions between temporal expectation and entrainment that reproduce aspects of human behavior and perception [11, 12]. But although the behavior of these models may be satisfying in certain applications, the groundwork underlying them is less so: key high-level concepts like the "attentional pulse" are difficult to define mechanistically or computationally, so the implementations of these concepts in models remain impressionistic.

An alternative approach to modeling the role of expectations in the brain is the "predictive processing" framework [13]. This framework posits that the brain engages in a continuous process of inferring the hidden causes of sensory events based on a learned understanding of how those causes produce sensation. Unlike the terms in Dynamic Attending Theory models, the terms in predictive processing models are directly linked to the formal inference problem being solved: the solution to the problem demands that certain quantities be computed, giving us reason to expect to find those quantities represented in the brain. In particular, "precision" or

certainty plays a key role, determining how new sensory information is weighted relative to existing beliefs about the hidden causes. According to this framework, estimates of hidden causes are inferred and updated through dynamic adjustments that minimize "prediction error" between a brain region's input and the input it is predicted to get based on current estimates, where distinct prediction errors are assigned relative weight by the precision of the associated predictions.

Here, we apply the predictive processing approach to the process of expectancy-based entrainment by formalizing it as an inference problem: namely, the problem of inferring the state of the exogenous process giving rise to a series of events in time. We use the mathematical tool of point processes to formulate a model of precise event timing. We derive an optimal solution to the inference problem, which we hypothesize corresponds with the brain's mechanisms for entrainment. The resulting models resemble Dynamic Attending Theory models, but introduce two key novel elements:

1. Dynamically estimated phase uncertainty moderates the balance between top-down and bottom-up influences on estimated phase.

2. Event expectations influence estimated phase even in the absence of actual events.

These elements allow them to reproduce aspects of human entrainment unaddressed by existing models, including:

1. Failure to track phase through excessive syncopation (events occurring at weakly expected times but omitted at strongly expected times).

2. Illusory contraction of intervals when expected events are omitted.

3. Near-linear corrections to phase after event timing perturbations, with larger (and even over-) corrections for stimulus trains with longer inter-onset intervals.

They are also significantly more flexible than Dynamic Attending Theory models in their descriptive power, allowing us to describe entrainment based on either periodic or aperiodic expectation patterns, and, as predictive processing models, they recast entrainment in a formal language that links it a the wide range of other cognitive phenomena.

In the next section, we formulate three versions of the problem of expectancy-based entrainment that are amenable to precise solutions, which we refer to collectively as the "phase inference framework." In the first, "Phase Inference from Point Process Event Timing" (PIPPET), a hidden phase variable advances steadily with added noise, and the observer is tasked with continuously inferring the phase based on the observation of events emitted probabilistically at certain phases with certain degrees of precision. In the second version, "Phase And Tempo Inference from Point Process Event Timing" (PATIPPET), the rate of phase advance (tempo) is also a dynamic variable with drift, and the solution simultaneously estimates phase, tempo, and certainty about both. The third version (mPIPPET) generalizes the first two to incorporate the observation of multiple types of events, each with distinct characteristic phases and precisions, into the inference process. We present variational filtering equations that approximate perfect Bayesian solutions to these problems.

In the Results section, we simulate these filters, drawing on music as a rich source of intuitive examples of entrainment informed by expectation. In doing so, we provide intuition into the range of behaviors of these solutions, and show how novel features introduced by the normative framework reproduce key aspects of human entrainment behavior that are not explained by other models. In the Discussion, we discuss the potential contributions of PIPPET and PATIPPET to the analysis of experimental data, to richer and more detailed models, and to our understanding of entrainment in the brain.

## Methods

Predictive processing should be a natural modeling framework for understanding rhythmic expectation and entrainment [14–16]. However, existing predictive coding models that operate in continuous time are structured to perform inference based on continuous observation, characterizing prediction errors in terms of deviation between a true level of input and a mean expected level [17, 18]. In other words, they describe predictions about "what" rather than "when." They are therefore ill-suited to characterizing moment-by-moment errors in *timing* prediction, which are made sporadically and separated by intervals mostly devoid of informative prediction error. This may be a fundamental shortcoming in modeling inference in the brain: behavior and neurophysiology suggests that information about "when" is carried by its own distinctive pathways and represented separately from "what," both in perceptual and motor tasks [6, 10, 19]. Bayesian methods have been applied to describe inferences about timing in the brain [20–22], but in these cases the problem the brain solves has been formulated as discrete inferences about consecutive intervals rather than a continuous inference process.

Here, we use event timing to inform a continuous variational inference process by first creating a generative model describing the probabilistic generation of precisely timed events and then variationally inverting that model. To model event generation, we use the mathematical tool of point processes.

### Phase inference from point process event timing (PIPPET)

PIPPET is the problem of dynamically estimating a hidden noisy phase variable based on the timing of events generated as a point process whose rate is modulated as a specific function of phase. The generative model consists of a phase $\phi \in \mathbb{R}$ that advances as a drift-diffusion process:

$$d\phi = dt + \sigma dW_t \tag{1}$$

and an inhomogeneous point process that generates events with probability $\lambda(\phi)$. This function is known to the observer. We will refer to $\lambda(\phi)$ as an "expectation template" because it describes the temporal structure of the observer's event expectations, though it can also be understood as a hazard rate for events. To achieve both analytical tractability and flexible descriptive power, we assume that $\lambda(\phi)$ is a sum of a constant $\lambda_0$ and a set of scaled Gaussian peaks indexed by $i = 1, 2, \ldots$ etc. Each Gaussian peak $i$ is centered at a mean phase $\phi_i$ with variance $v_i$ and scale $\lambda_i$:

$$\lambda(\phi) = \lambda_0 + \sum_i \lambda_i \varphi(\phi | \phi_i, v_i) \tag{2}$$

where $\varphi(\cdot | m, v)$ denotes the pdf of a Gaussian distribution with mean $m$ and variance $v$.

- Each Gaussian mean $\phi_i$ represents a phase at which an event is expected;
- $\lambda_i$ represents the strength of that expectation;
- and $v_i^{-1}$ is the temporal precision of that expectation.
- $\lambda_0 > 0$ represents the rate of events being generated as part of a uniform noise background unrelated to phase.

The point process with rate described by (2) can be understood as a sum of independent point processes $i$, one for each expectation peak and one for the uniform background process with rate $\lambda_0$, whose events are indistinguishable. The mathematics of updating a phase estimate

at an event can be understood to involve a causal inference on which of these processes caused each event.

$\lambda(\phi)dt$ is the likelihood function over $\phi$ associated with the occurrence of an event, so $\lambda(\phi)$ is a rescaled likelihood function. See Fig 1A for illustration.

Note that $\phi$ is assumed to be on the real line, not the circle. This design decision allows PIP-PET to entrain to temporally patterned expectations with or without periodic structure by choosing a periodic or aperiodic expectation template $\lambda$.

Given a series of event times $\{t_n\}$ tallied by an event-counting function $N_t : \mathbb{R} \rightarrow \mathbb{Z}^{0+}$, an expectation template $\lambda(\phi)$, and a prior distribution $p_0(\phi)$ describing the distribution of phase at time $t = 0$, the observer's goal is to infer a posterior distribution $p_t(\phi) = p(\phi|N_{\tau<t})$ describing an estimate of phase $\phi$ at any time $t$ based on the event history up to $t$.

In [23], Snyder derives an exact PDE for the evolution of this posterior distribution over time. Following the predictive processing ansatz of maintaining Gaussian posterior distributions (the Laplace assumption), which provides both computational tractability and neurophysiological plausibility by reducing the representation of the posterior to a mean and a variance, we project the posterior onto a Gaussian at each $dt$ time-step. We do this by moment-matching: we use Snyder's solution to determine the evolution of the mean and variance of the posterior, and then replace the true posterior with a Gaussian with the same mean and variance. This choice of Gaussian is the choice with minimum KL divergence from the true posterior [24], and therefore also minimizes the free energy of the solution within the family of possible Gaussian posteriors in accordance with the Free Energy Principle [25].

The result of this derivation is a generalization of a Kalman-Bucy filter with Poisson observation noise. Eden and Brown [26] have derived an explicit form for this filter, but it relies on a local approximation of the rate function $\lambda$ that hides some of the interesting effects of events expected at nearby time points. For $\lambda$ a mixture of Gaussians, we derive a filter directly from Snyder's solution in [23] that more accurately approximates the optimal (Bayesian) solution. The derivation is presented in S2 Text.

**Solution: The PIPPET filter.** At any time $t$, let $\mu_t$ denote the mean and $V_t$ denote the variance of the Gaussian posterior. At each event time $t$, we let $\mu_t$ and $V_t$ equal the left-hand limits of $\mu$ and $V$ before the event, and we write $\mu_{t+}$ and $V_{t+}$ to denote their right-hand limit values after the event ($\mu$ and $V$ are left-continuous). Let $dN_t$ denote the increment in the event-counting process at time $t$, which is either 0 or 1 with probability one. $\mu_t$ and $V_t$ evolve according to the stochastic differential equation:

$$\begin{cases} d\mu = & dt + (\hat{\mu} - \mu)(dN_t - \Lambda dt) \\ dV = & \sigma^2 dt + (\hat{V} - V)(dN_t - \Lambda dt) \end{cases} \qquad (3)$$

or, equivalently, they evolve between events according to the ODE: $\begin{cases} \dot{\mu} = & 1 - \Lambda(\hat{\mu} - \mu) \\ \dot{V} = & \sigma^2 - \Lambda(\hat{V} - V) \end{cases}$

and reset at each event to $\mu_{t+} = \hat{\mu}$ and $V_{t+} = \hat{V}$, where we define

$$\hat{\mu} := \frac{\lambda_0}{\Lambda}\mu_t + \sum_{i=1,\dots} \frac{\Lambda_i}{\Lambda}\hat{\mu}_i$$

$$\hat{V} := \frac{\lambda_0}{\Lambda}\left(V_t + (\mu_t - \mu_{t+})^2\right) + \sum_{i=1,\dots} \frac{\Lambda_i}{\Lambda}\left(\hat{V}_i + (\hat{\mu}_i - \mu_{t+})^2\right)$$

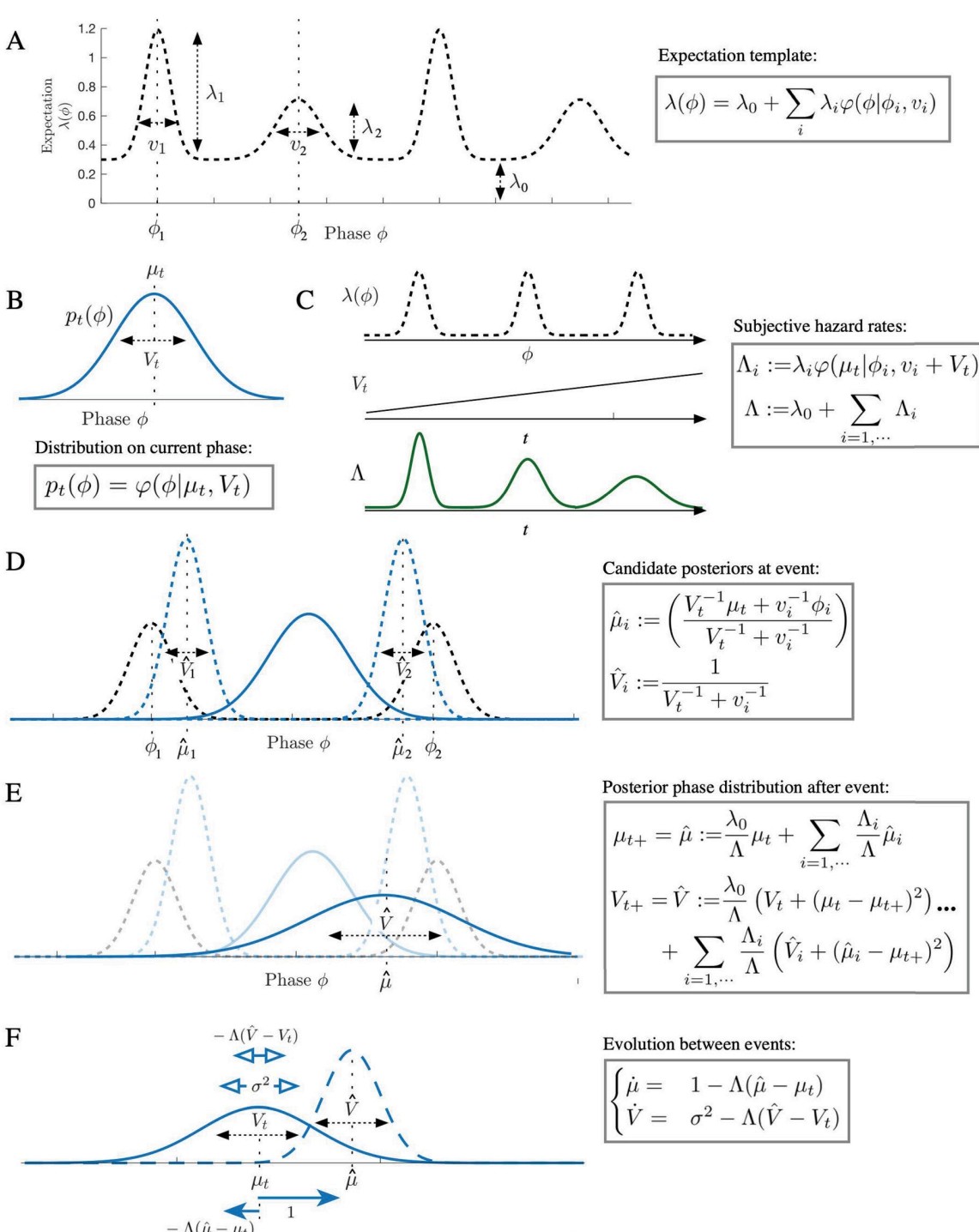

**Fig 1. Illustration of the PIPPET filter.** A) In the PIPPET generative model, $\lambda(\phi)$ represents the instantaneous rate of events occurring when the underlying temporal process is at phase $\phi$. This is assumed to be a sum of Gaussian-shaped functions with means $\phi_i$ representing the phases at which specific events are expected, variances $v_i$ representing (the inverse of) the temporal precision of the expectations, and scales $\lambda_i$ representing the strength of the expectations. A constant $\lambda_0$ is also added, representing the instantaneous rate of events unrelated to phase. B) At any time $t$, the filter's estimate of current phase $p_t(\phi)$ is forced to be a Gaussian with mean $\mu_t$ (the estimated phase at time $t$) and variance $V_t$ (the level of uncertainty about the phase estimate). C) These allow us to define a subjective hazard rate $\Lambda$ (implicitly a function of time) representing the degree to which an event is anticipated at $t$, and conditional subjective hazard rates $\Lambda_i$ representing the degree to which an event is anticipated from peak $i$. These hazard rates become less precise as phase uncertainty $V_t$ increases. D) Each peak $i$ of $\lambda$ is associated with a "candidate posterior" with mean $\hat{\mu}_i$ and variance $\hat{V}_i$—this would be the

posterior on phase if the event were known to come from peak $i$. E) At an event, the phase distribution resets to a Gaussian with mean $\hat{\mu}$ and variance $\hat{V}$. These incorporate the influences of each candidate posterior, and $\hat{V}$ can increase if the cause of the event is ambiguous (as dramatically illustrated above). F) Between events, $\mu_t$ increases at rate 1 and $V_t$ grows at rate $\sigma^2$. Additionally, $\mu$ and $V$ are pushed away from $\hat{\mu}$ and $\hat{V}$ with a strength proportionate to subjective hazard rate $\Lambda$.

(Note that in this formulation, $\mu_{t+}$ must be calculated before $V_{t+}$).

$$\hat{\mu}_i := \frac{V_t^{-1}\mu_t + v_i^{-1}\phi_i}{V_t^{-1} + v_i^{-1}} \quad \text{and} \quad \hat{V}_i := \frac{1}{V_t^{-1} + v_i^{-1}}$$

$$\Lambda_i := \lambda_i \varphi(\mu_t | \phi_i, v_i + V_t) \quad \text{and} \quad \Lambda := \sum_i \Lambda_i$$

These terms are illustrated in Fig 1. Intuitively,

- $\Lambda$ (implicitly a function of $\mu_t$ and $V_t$) is the degree to which an event is anticipated at $t$ while taking into account uncertainty about underlying phase, also known as the "subjective hazard rate". $\Lambda_i$ is the degree to which an event is anticipated from peak $i$ (the "conditional subjective hazard rate").

- At each event time $t$, $\lambda(\phi)$ serves as a (rescaled) likelihood function for phase, and the role of prior is played by the phase distribution $p_t$, a Gaussian with mean $\mu_t$ and variance $V_t$. Each peak $i$ of $\lambda$ is a possible "cause" of the event, as is the background event rate $\lambda_0$. Each peak is associated with a "candidate posterior" with mean $\hat{\mu}_i$ and variance $\hat{V}_i$—this would be the posterior on phase if the event were known to be caused by peak $i$. $\hat{\mu}_i$ is a weighted sum of the current mean estimated phase $\mu_t$ and the center $\phi_i$ of expectation peak $i$, weighted by their respective precisions. Note that, following the predictive processing ansatz, this is the phase that minimizes precision-weighted prediction error with respect to predicted event timing and predicted phase.

- At an event, the phase distribution resets to a Gaussian with mean $\hat{\mu}$ and variance $\hat{V}$. These are weighted sums of the influences of each candidate posterior, each weighted by conditional subjective hazard rate $\Lambda_i$. The expression for $\hat{V}$ contains additional terms $(\hat{\mu}_i - \mu_{t+})^2$ and $(\mu_t - \mu_{t+})^2$, which cause the variance of the posterior to increase if the cause of the event is ambiguous.

- The background rate $\lambda_0$ acts as an alternative possible cause for any event. It serves to weight the posterior phase distribution toward the prior distribution before the event, and gives rise to causal ambiguity for any event and a resulting increase in posterior variance.

- Between events, each $dt$ time step is taken as a Bayesian inference with likelihood $1-\lambda(\phi)dt$ and with a Gaussian prior consisting of the posterior of the previous time step carried forward by $dt$ according to the Fokker-Planck evolution associated with Eq (1). This prior causes $\mu_t$ to increase steadily and $V_t$ to grow at rate $\sigma^2$. The likelihood pushes $\mu$ and $V$ away from $\hat{\mu}$ and $\hat{V}$ with a strength proportionate to subjective hazard rate $\Lambda$. Thus, the absence of an event continuously pushes the posterior in the opposite direction as would the occurrence of an event.

## Phase and tempo inference from point process event timing (PATIPPET)

PATIPPET extends PIPPET by making the rate of phase advancement itself a noisy dynamic variable subject to ongoing inference. The dynamic state of the system is now a two-

dimensional vector $\mathbf{x} = \begin{pmatrix} \phi \\ \theta \end{pmatrix}$, where $\phi$ is the phase as above, $\theta$ is the rate of phase advancement (or tempo), and $\sigma$ and $\sigma_\theta$ are the levels of phase and tempo noise, respectively:

$$d\mathbf{x} = \begin{pmatrix} \theta \\ 0 \end{pmatrix} dt + \begin{pmatrix} \sigma dW_t \\ \sigma_\theta dW_t^\theta \end{pmatrix} \tag{4}$$

As above, an inhomogeneous point process generates events with probability based on an expectation template $\lambda$, which in this case is a function of both phase $\phi$ and tempo $\theta$. In this formulation, we want events to occur with a certain probability in each $d\phi$ phase bin regardless of tempo, which we can accomplish by scaling the event rate by $\theta$:

$$\lambda(\phi, \theta) = \theta \left( \lambda_0 + \sum_i \lambda_i \varphi(\phi | \phi_i, v_i) \right) \tag{5}$$

Note that this is the same as the PIPPET expression for event rate if we set $\theta = 1$.

As before, the observer's goal is to infer a posterior distribution at any time $t$ using preceding event times; now the distribution $p_t(\mathbf{x})$ describes an estimate of both phase and tempo. A similar derivation provides a point-process Kalman-Bucy filter that optimally serves this function within the constraint of Gaussian posteriors, providing a running estimate of a mean phase and tempo $\boldsymbol{\mu}_t$ and a phase/tempo covariance matrix $\mathbf{V}_t$. The solution is presented in S1 Text and its derivation is presented in S2 Text.

The resulting PATIPPET filter generalizes the PIPPET filter, and is identical if the initial tempo distribution is set to a delta distribution at $\theta = 1$ and $\sigma_\theta$ is set to zero. At each event, the distribution of phase and tempo is discontinuously updated to a 2D Gaussian posterior, which evolves continuously between events. This scheme is similar to [27], which estimates phase and tempo by updating a 2D Gaussian posterior, but is updated in continuous time and is significantly more flexible in its capacity to track phase based on arbitrary expectation templates.

## PIPPET with multiple event streams (mPIPPET)

Finally, we generalize PIPPET to include multiple types of events (indexed by $j$), each generated as point processes with rates determined by functions $\lambda^j(\phi)$ of a single underlying phase:

$$d\phi = dt + \sigma dW_t \tag{6}$$

$$\lambda^j(\phi) = \lambda_0^j + \sum_i \lambda_i^j \varphi(\phi | \phi_i^j, v_i^j) \tag{7}$$

The Kalman-Bucy estimate of phase for this model is described by mean $\mu$ and variance $V$ evolving according to the ODE

$$\begin{cases} \dot{\mu} = & 1 - \sum_j \Lambda^j (\hat{\mu}^j - \mu) \\ \dot{V} = & \sigma^2 - \sum_j \Lambda^j (\hat{V}^j - V) \end{cases} \tag{8}$$

and resetting to $\mu_{t+} = \hat{\mu}^j$ and $V_{t+} = \hat{V}^j$ when an event occurs in stream $j$, where we define $\Lambda^j$, $\hat{\mu}^j$, and $\hat{V}^j$ as we defined $\Lambda$, $\hat{\mu}$, and $\hat{V}$ above but in reference only to event stream $j$.

The same adjustment can be made to the PATIPPET generative model, and the PATIPPET filter can be similarly generalized to account for multiple event streams.

## Computational simulations

In the "Results" section, we conduct a series of simulations to illustrate how the novel terms representing dynamic tracking of uncertainty and the influence of expectations in the absence of events allow the PIPPET and PATIPPET filters to reproduce perceptual and behavioral observations during human entrainment to auditory rhythms. Parameters for these simulations are listed in S3 Text. Simulations were conducted in MATLAB [28] using code available at https://github.com/joncannon/PIPPET.

# Results

## Updating posterior in response to events

We simulated the PIPPET filter with a single expectation peak and varied parameters to illustrate its basic behavior (Fig 2). Fig 2 column $i$ illustrates the effect of an event on the phase estimate as a function of initial estimated phase $\mu_t$. Events occurring when $\mu_t$ is near an expected event phase $\phi_1$ caused $\mu$ to shift linearly toward $\phi_1$. When we set the uniform rate of background events $\lambda_0 > 0$, events occurring far from the expected event phase $\phi_1$ were attributed to the background and therefore caused negligible adjustment to the phase estimate. Phase uncertainty $V_t$ decreased at events except when $\lambda_0$ was positive and $\mu$ was not sufficiently close to $\phi_1$; in this case, $V_t$ increased due to causal ambiguity, or stayed the same if the cause was unambiguously the uniform background source.

## Tracking complex rhythms with uneven subdivision

The PIPPET framework describes entrainment to rhythms in which each expected event phase may or may not be populated by an event. It is formulated in sufficient generality to describe entrainment to rhythms based on timing expectations with complex, non-isochronous stress patterns [29] and with non-integer duration ratios using suitably constructed (presumably learned) expectation templates $\lambda(\phi)$. Such rhythmic patterns have been shown to support highly precise synchronization in musicians with appropriate training and enculturated expectations [30], and should therefore be accounted for by models of human entrainment.

As an example of entrainment to a complex rhythm based on a temporal structure with non-integer duration ratios, we simulated entrainment to a swing rhythm. The rhythm is based on an underlying grid of "swung" eighth notes, where the first event of every pair is followed by a slightly longer inter-event duration than the second. Though the "swing" feel is often caricatured using eighth note pairs with a 2:1 duration ratio, this value has been shown to vary by with style and tempo and is certainly not limited to small integer ratios [31]. We used an expectation template with a swing ratio of 3:2 (though the exact ratio is not important) and associated the first eighth note in each pair with a stronger expectation than the second. The PIPET filter entrained to a complex, syncopated rhythm based on this template, drawing on the timing of both strongly and weakly expected events (Fig 3A). It corrected its phase estimate when an event timing shift or a phase shift was introduced into the rhythm (Fig 3B and 3C).

## Failure mode: Too much syncopation

The phase inference framework can account for human failures to track perfectly timed rhythms, i.e., rhythms in which every event falls at a peak of the expectation template. A prime example of this failure mode in human rhythm tracking is tracking overly syncopated rhythms (rhythms with a predominance of events at time points with weaker expectations). Listeners

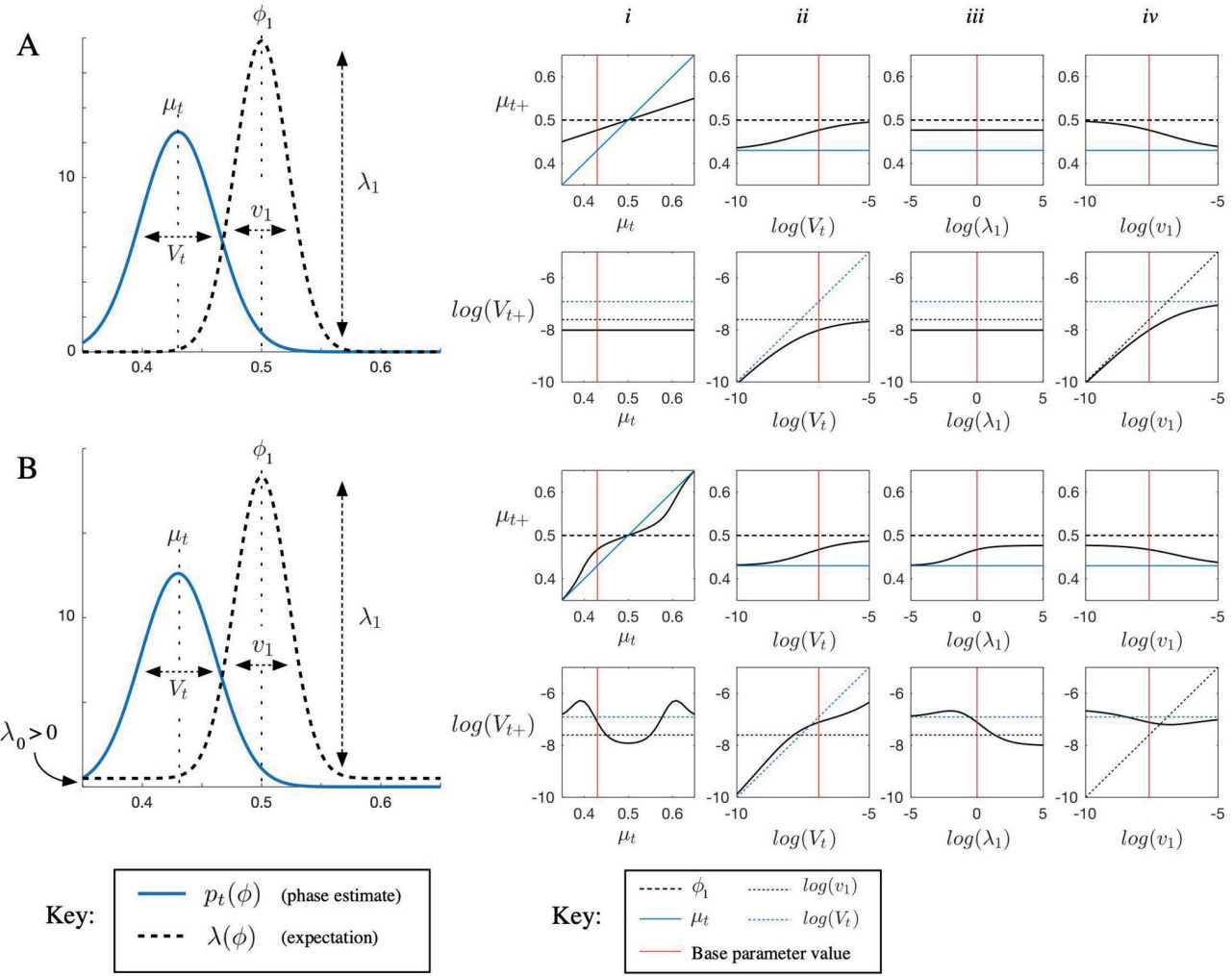

**Fig 2. Characterizing PIPPET's behavior at events.** A) An event is expected at phase $\phi_1 = 0.5$ with variance $v_1$ and expectation strength $\lambda_1$. The expected background event rate is set to $\lambda_0 = 0$. An event occurs when the phase estimate is at $\mu_t$ with uncertainty $V_t$. Panels in columns *i-iv* show the resulting mean $\mu_{t+}$ and variance $V_{t+}$ of the posterior on phase as the parameters $\mu_t$, $V_t$, $\lambda_1$, and $v_1$ are varied. *i*) $\mu$ is corrected linearly toward $\phi_1$, while $V$ decreases uniformly regardless of initial phase. *ii*) Corrections to $\mu$ are more thorough when $V_t$ is large. *iii*) These corrections do not depend on $\lambda_1$. *iv*) These corrections are more thorough for smaller $v_1$. B) The same simulations are carried out with background event rate $\lambda_0 = 0.5$. *i*) If $\mu_t$ is close to $\phi_1$, it is linearly corrected toward $\phi_1$ and $V_t$ decreases; if it is far, no correction is made. In the liminal zone, $V_t$ increases due to the ambiguity of whether the event was related to the expectation peak or due to the background source. *ii*) $V_{t+}$ is larger due to the effect of ambiguity as to whether the event is associated with $\phi_1$ or with the background rate. *iii*) Now the correction depends on $\lambda_1$: stronger expectations make this peak the favored cause relative to the background source. *iv*) Note that if the expectation peak is extremely narrow, $V_{t+}$ may still be large after the event and $\mu_t$ may not fully reset to $\phi_1$ due to the aforementioned causal ambiguity.

tend to "re-hear" such rhythms by attributing events to metrical positions where events are more strongly expected [32, 33].

In PIPPET, these failures consist of inferring the presence of phase noise where none actually occurred. Such behavior is a necessary consequence of Bayesian optimality: a given stimulus may be generated by different combinations of phase noise and point process event generation noise, and the inference process is concerned only with the most likely explanation for the stimulus, which may include phase noise even if the stimulus was actually generated without it.

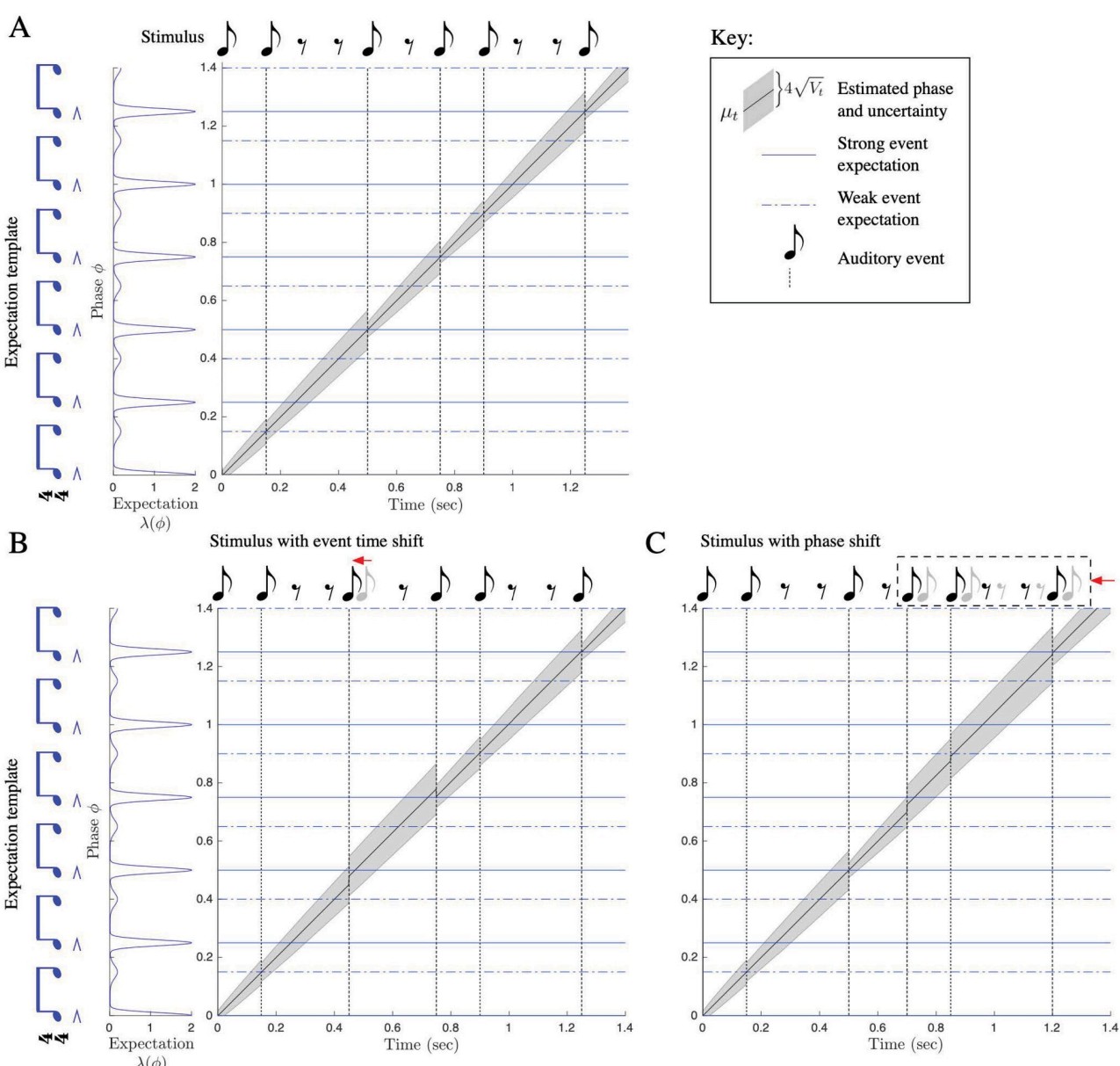

**Fig 3. Tracking phase through swung rhythms.** PIPPET is given a pattern of expectations representing "swung" eighth notes, with alternating longer and shorter inter-event durations and stronger, more precise expectations on the first of every pair. Dotted lines correspond to weaker expectations and solid lines correspond to stronger expectations. A) Phase is successfully tracked over the course of a rhythmic stimulus, with phase uncertainty growing between events and contracting at events. B) One event in the rhythm is shifted earlier in time. Estimated phase $\mu_t$ adjusts partially to compensate for the timing shift, and then adjusts back at the subsequent event. Uncertainty $V_t$ is not as effectively reined in by these unpredictably-timed events, but decreases as later events corroborate the corrected phase estimate. C) A phase shift is introduced into the rhythm, moving all subsequent events earlier in time. When the first early event arrives, uncertainty increases. Estimated phase is corrected over the first few events after the shift, and $V_t$ decreases most substantially when the estimate $\mu_t$ is corroborated by a strongly expected event happening at the appropriate estimated phase.

Using the expectation template with a swing grid as in the previous section, we simulated a strongly syncopated rhythm (Fig 4A). The rhythm's phase was not tracked successfully due to a convergence of two factors: the disproportionate influence of the higher peaks of the expectation template, and the accumulation of phase uncertainty $V_t$. Phase uncertainty was only slightly reduced by events occurring at weakly expected phases, so it accumulated over the

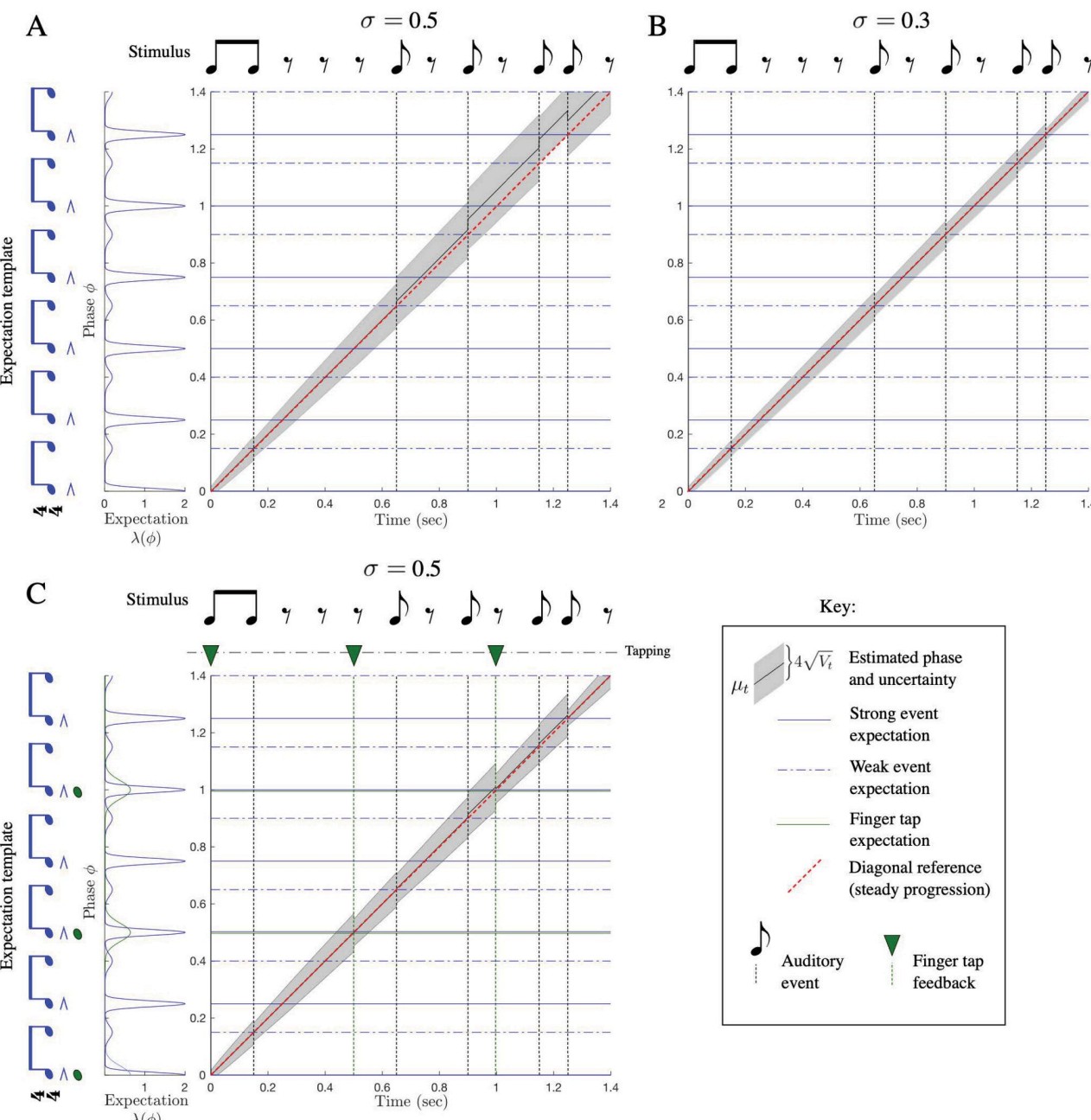

**Fig 4. Too much syncopation causes rhythm tracking failure.** A predominance of events associated with weak expectations combined with accumulated phase uncertainty can lead to a failure to track phase accurately. A) In this example, phase uncertainty $V$ increases over a long silence. At the next event, this high uncertainty leads the model to partially attribute a weakly expected event to the nearby phase at which an event is strongly expected. As a result, the model ends up aligning the fifth event with a strong phase rather than a weak one, and overestimating phase at the final event (correct phase marked with yellow dot). B) When the rate of accumulation of phase uncertainty (i.e., the expected phase noise $\sigma^2$) is decreased, phase is tracked correctly. C) Alternatively, phase can be tracked successfully by inserting an isochronous stream of finger taps and a suitable template for the alignment between expected auditory feedback from the taps and phase. We use mPIPPET to simulate an expectation for isochronous taps (green notes and trace on the left). For simplicity, taps are placed every 0.5 sec; however, even noisy taps generated based on estimated phase could serve to reduce phase uncertainty and avoid a total phase tracking failure.

course of the rhythm, and especially during the long silence. Once $V_t$ was large, indicating the possibility of substantial phase noise having accumulated, the higher expectation peaks $\phi_i$ became the most likely explanations for events that were actually perfectly timed to coincide with nearby lower peaks—since precise event timing was no longer a reliable indicator of the source of an event, local peak height became the best indicator, and higher peaks won out. Thus, at each event, the estimated phase was adjusted to better align the higher peaks with the events.

The same rhythm could be successfully tracked in two alternate conditions. First, it was successfully tracked when we decreasing the rate of accumulation of phase uncertainty $\sigma^2$ (Fig 4B), demonstrating the key role of uncertainty in making the system susceptible to the disruptive effect of syncopation. Second, it was successfully tracked when an additional stream of sensory input was added by simulating an isochronous finger tap (Fig 4C). We used mPIPPET to create a second expectation template for tapping. As phase tracking was simulated, we planned new tap events just before $\mu$ reached expected tap phases by extrapolating $\mu$ forward. When taps occurred, phase uncertainty decreased, reducing the disruptive effects of syncopation. Note that planning actions specifically to fulfill sensory expectations and using this sensory feedback to inform inference about the outside world is an example of "active inference", the principle framework for understanding action in the literature on predictive processing [25].

## Tempo inference

We simulated the PATIPPET filter with basic metronomic expectations to observe its capacity to infer phase and tempo at once. We gave the model a wide initial range of possible tempi and a simple metronomic stimulus with actual tempo near the upper end of that range. In these conditions and with the parameter set we chose, the model established the appropriate tempo and phase to within a tight range over the course of the first two events (Fig 5).

In addition to its value as a model of human rhythmic cognition, the PATIPPET filter shows promise as a general-purpose tempo tracking algorithm for musical applications. This would require a principled method of choosing values for the various free parameters of the generative model, which might be done a priori based on a labeled corpus, adaptively over the course of listening, or through some combination of the two. We leave a more thorough exploration of the relative performance of this model to future work.

## Period-dependent corrections

In sensorimotor entrainment literature, finger taps entrained to a metronome generally shift to correct a certain fraction of an event timing perturbation on the next tap. This fraction is called $\alpha$. In human subjects, $\alpha$ has repeatedly been observed to increase linearly with metronome period ("inter-onset interval," or IOI), exceeding 1 (i.e., over-correction) for sufficiently long IOIs [34, 35].

The phase inference framework offers a principled explanation for $\alpha$ increasing with IOI. During an event-free interval, phase uncertainty increases over time. When an event does occur, the precision of the prior distribution on phase and tempo is weighed against the precision of the likelihood function associated with the expectation of that event. If the prior is less precise due to accumulated uncertainty, the precision of the likelihood weighs more heavily against it and the adjustment in phase is more thorough. Thus, all else being equal, events spaced more widely apart in time induce more extensive phase corrections.

Since the strongest phase correction PIPPET can make at an event is to fully update the phase estimate to the expected event time, it cannot account for $\alpha$ values above 1. However, it

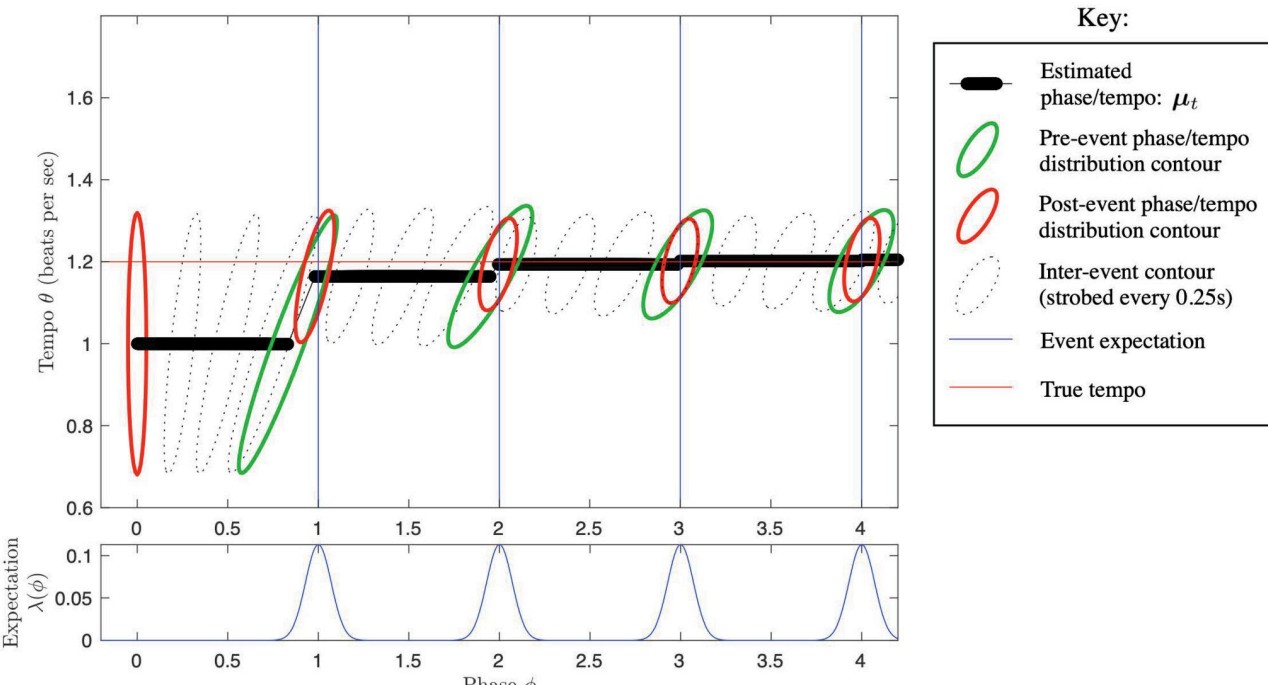

**Fig 5. The PATIPPET filter estimates phase and tempo.** PATIPPET is initialized with high tempo uncertainty. The first event occurs relatively early, causing the estimated tempo to increase. Each subsequent event occurs close to the time expected based on the estimated phase and tempo, causing the posterior to contract in both the phase and tempo direction as its prediction of event time is fulfilled and its phase and tempo estimates are corroborated. Ultimately, PATIPPET settles on a narrow distribution around the appropriate tempo as it continues to accurately estimate phase.

has been previously suggested that $\alpha$ may exceed 1 for long metronome periods due to some period correction occurring in addition to phase correction [34]. We were therefore curious to see whether PATIPPET could reproduce the linear increase of $\alpha$ with increasing IOI up to and beyond $\alpha = 1$.

In Fig 6, we show that with appropriate parameters, PATIPPET can indeed reproduce the experimental observation of a near-linear increase in $\alpha$ from below to above 1 as IOI increases. In PATIPPET, this phenomenon is a natural consequence of optimal inference in the context of phase and tempo uncertainty that accumulates between observed events.

## Time warping in the absence of expected events

When an event in a rhythmic stimulus is strongly expected but no event occurs, an optimal Bayesian observer should initially be biased to believe that in spite of their current phase estimate, the stimulus may not have reached the expected event phase yet. The result should be that a perfectly timed event later in the stimulus will seem to be arriving earlier than expected: in other words, the tempo of the stimulus will seem to accelerate. The degree of this effect will depend on the observer's degree of phase and tempo uncertainty.

There is evidence of such an effect in human rhythm perception. The "filled duration" illusion is the impression that an isochronous sequence has changed tempo when it is initially subdivided by additional predictable events and then subdivisions are eliminated. According to multiple reports, the magnitude of this effect is reduced or eliminated if the empty intervals precede the filled intervals [36–39] (though there is some disagreement about this [40]), suggesting that it is indeed the established expectation of continuing subdivision that interferes with the perceived passage of time when the subdivisions cease. A second result that could be

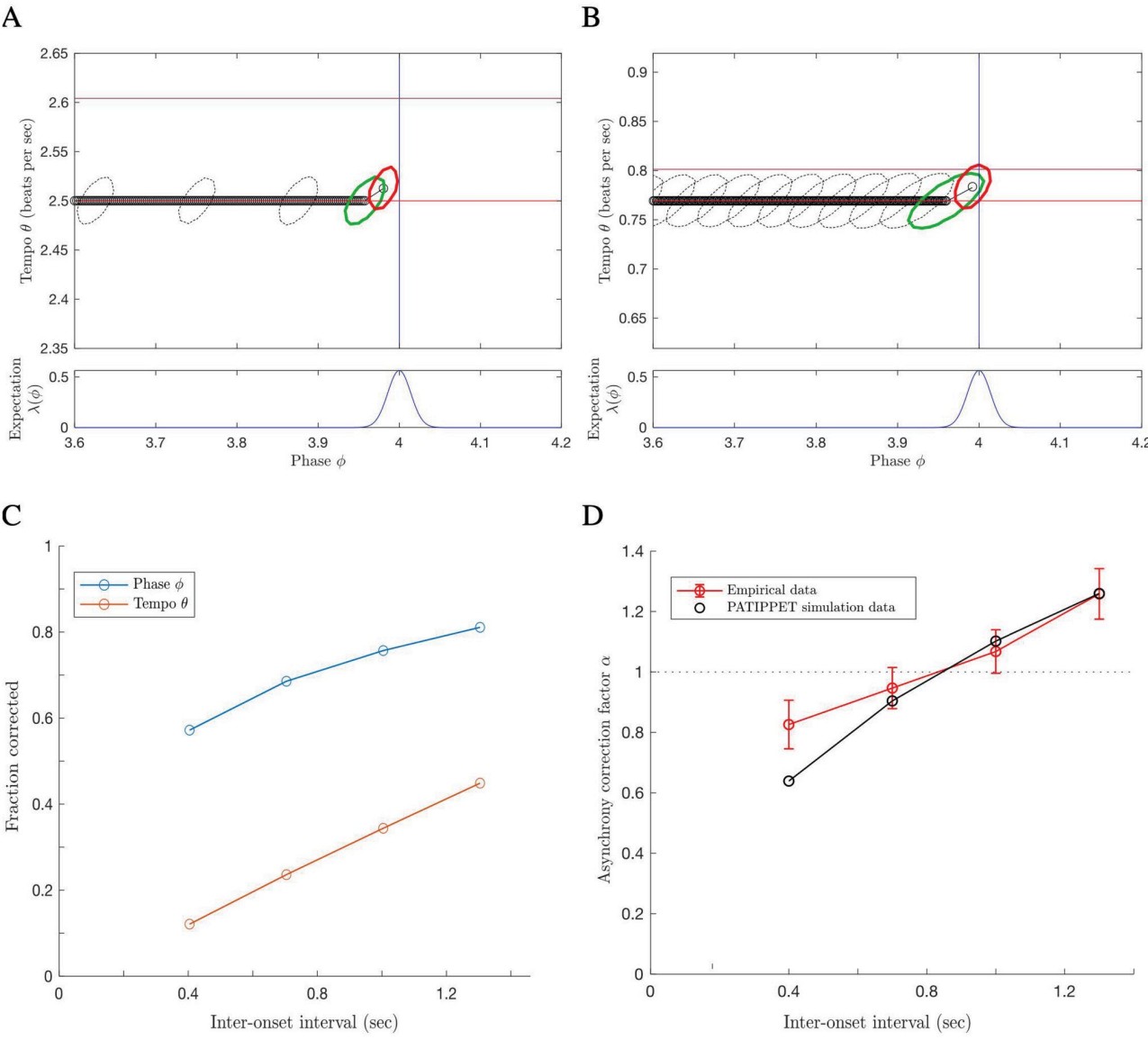

**Fig 6. PATIPPET reproduces human tapping data showing stronger error correction for longer inter-onset intervals.** A and B) The distribution on phase and tempo leading up to and following a phase shift at the fourth event in an isochronous sequence for two different metronome tempi, i.e., two different inter-onset intervals. (Same color key as Fig 5, but with phase/tempo distribution contours strobed every .05 sec). Note that when the IOI is short, PATIPPET arrives at the phase-shifted event with a high degree of phase and tempo certainty. C) PATIPPET makes a proportionally larger correction to phase and tempo for long IOIs than for short IOIs due to the greater degree of uncertainty preceding each event. D) Alpha ($\alpha$) is the proportion of a phase shift that is corrected at the next tap time. With this set of parameters, PATIPPET reproduces the empirical observation from [35] that the phase shift is undercorrected when IOIs are short and overcorrected $\alpha > 1$ when IOIs are long.

similarly accounted for is the surprising finding in [41] that a participant tapping along with a subdivided beat delays their tap following the omission of an expected subdivision. If taps are planned to coincide with the arrival of a specific mean estimated phase, then the slowing of estimated phase induced by an omission of a strongly expected event should indeed delay the subsequent tap.

We stimulated PATIPPET with a strong isochronous expectation template by scaling up λ and presented it with a "filled duration" in which all expected events occurred and an "empty duration" in which events occurred only at the beginning and end of the interval (Fig 7).

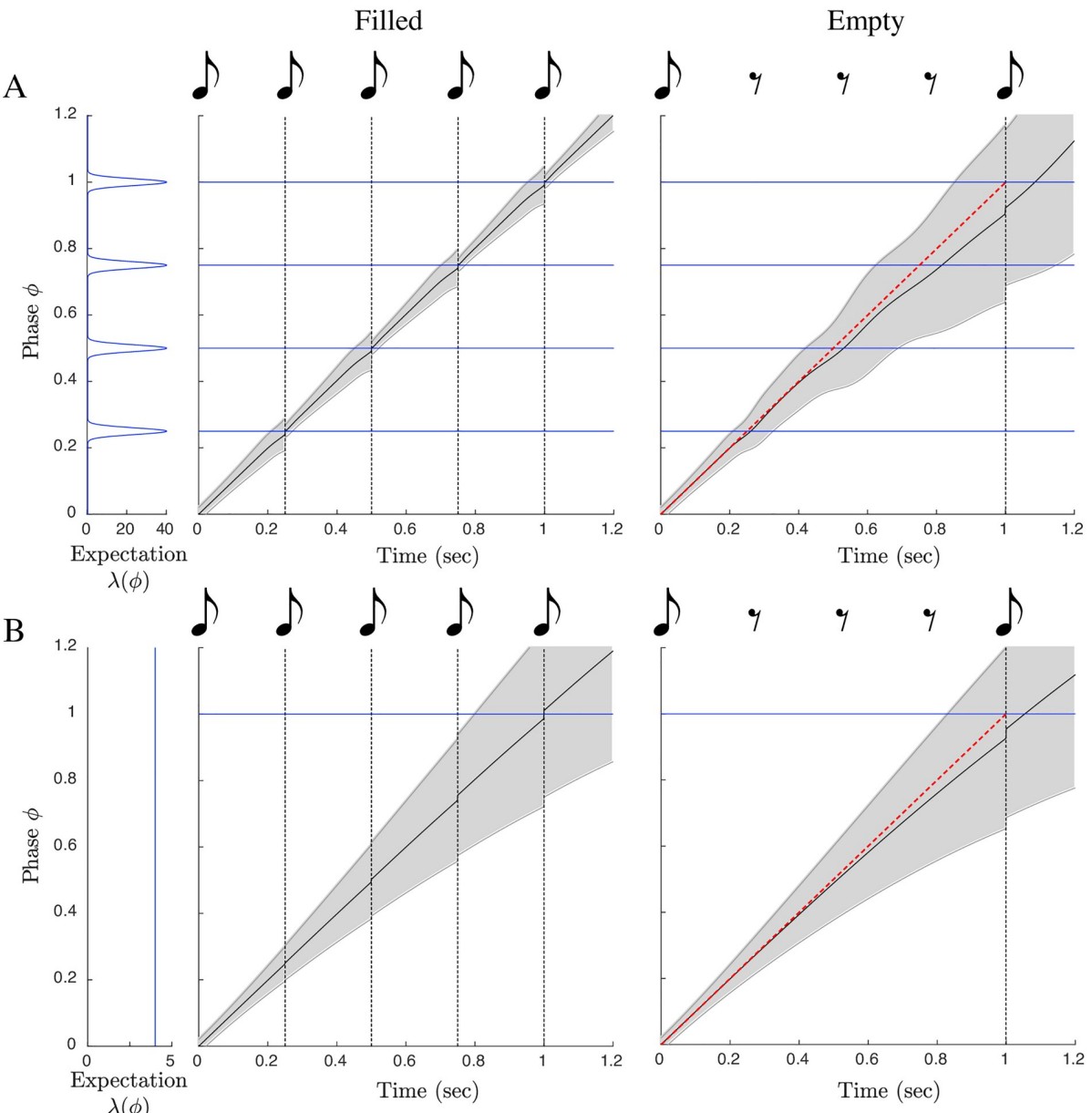

**Fig 7. The filled duration illusion: Time warping by the omission of strongly expected events.** (Same image key as 4, with shading displaying PATIPPET phase variance.) A) PATIPPET is simulated with strong expectations for isochronous events. Left: When a set of strongly expected events occur as expected (a filled duration), estimated phase stays on track, advancing (on average) at a rate of 1. Right: When the duration is empty, estimated phase deviates from steady progression (red diagonal) by dragging as each expected event point approaches and passes, leading to the illusion that the event marking the end of the interval has arrived earlier than expected. B) PATIPPET is simulated with a high expected background rate of events $\lambda_0$, but no phase-specific event expectations $\phi_i$. In this case, too, an empty duration leads to dragging estimated phase and an unexpectedly early final event.

PATIPPET loyally tracked phase through the filled duration; however, when strongly expected events were omitted, the mean phase estimate slowed down at each expected event phase, leading to an overall slowing in estimated phase advance and an unexpectedly early onset of the event marking the end of the empty duration (Fig 7A).

Specifically timed event expectations are not necessary to produce a filled duration illusion: random raindrop sounds were sufficient to lengthen produced intervals during audiomotor

synchronization task [42]. In PATIPPET, a filled duration effect was also produced when the expectation template consisted only of a high expected background rate of events $\lambda_0$. In this case, estimated phase advance slowed during the empty interval because estimated tempo dropped. The PATIPPET filter effectively noted that not as many events were occurring as expected, and in response it lowered estimated tempo because a lower event rate is expected at a lower tempo. This type of explanation could be invoked to offer a normative account for other non-rhythmic filled interval illusions, though doing so is beyond the scope of this work.

## Discussion

Here were have presented PIPPET, a framework representing entrainment to a time series of discrete events based on a template of temporal expectations. PIPPET treats the event stream as the output of a point process modulated by the state of a hidden phase variable. The PIPPET filter uses variational Bayes to continuously estimate phase and track phase uncertainty based on this generative model. PATIPPET extends PIPPET to include a generative model of tempo change, and the PATIPPET filter simultaneously estimates phase, tempo, and the covariance matrix representing their uncertainty and their codependence. This framework is intended to serve as a hypothesis for how the human brain integrates auditory event timing to inform and update an estimate of the state and rate of an underlying temporal process.

PIPPET and PATIPPET reproduce several qualitative features of human entrainment, including realistic failures to track overly perfectly-timed but over-syncopated rhythms, perceived acceleration of a metronomic pulse when strongly expected events are omitted, and error correction after metronome timing perturbations that increases with increasing inter-onset interval. We show that these three phenomena all follow naturally from our framing of entrainment as a process of Bayesian inference based on specific phase-based temporal expectations.

### Relationship to other models of timing

The dynamics of PIPPET and PATIPPET in response to sensory events are similar to dynamics of other entrainment models that correct phase and period based on event timing, e.g., [43, 44]. Models based on Dynamic Attending Theory, e.g., [11, 12], are also similar in explicitly modeling timing expectations and their effect on phase and period adjustment. The phase inference framework differ from these existing models in four key ways. First, they are derived as optimal solutions to specific inference problems, and therefore all modeling decisions can be justified within a normative framework. Second, they are formulated in sufficient generality to describe entrainment based on non-isochronous and even aperiodic temporal expectations, an area that has lately received increasing experimental attention [6, 45, 46] but has been largely neglected in entrainment modeling. Third, they allow expectations to influence the inferred phase even in the absence of sensory events, creating the time-warping effect of disappointed expectations evidenced in humans by the "filled duration" illusion. Finally and most critically, they explicitly track uncertainty in phase and tempo, providing a system for moderating between assimilation of new timing data and loyalty to an internal sense of time.

Bayesian methods have been used elsewhere to analyze rhythmic structure as time series of point events. Some of these are application-focused methods that require offline analyses [47, 48] and therefore do not serve as satisfying models of real-time behavior. Cemgil et al (2000) [27] use a Kalman filter that tracks a distribution on phase and tempo similarly to PATIPPET. However, this model is structured to infer phase and tempo event-by-event rather than in continuous time, and is not equipped to handle complex rhythms or temporal structures more complex than approximate isochrony.

Bayesian inference has also been used to model timing estimation in the brain (e.g., [20, 21]), but it is generally used to describe inferences about discrete variables like interval durations and event times, whereas PIPPET describes a continuous inference process underlying predictions about event times. One such model leading to particularly PIPPET-like results was presented in Elliot et al 2014 [22]. The authors created a Bayesian model to explain the results of an experiment that had participants tap along to a stimulus consisting of two jittered metronomes. The model behaves similarly to PIPPET in that it estimates the next event time using a weighted average of previous event times and prior beliefs, with weights informed by expected timing precision. However, like [27], their model infers the anticipated timing of discrete, metronomic events, whereas PIPPET predicts and updates an underlying phase in continuous time and can therefore generalize to non-isochronous and complex rhythms and account for the effects of event omissions. Additionally, in order to account for participants ignoring events far from predicted time points, they introduce the assumption that participants repeatedly test the hypotheses that events come from one or two separate streams, whereas PIPPET naturally accounts for this phenomenon by attributing stray events to a uniform background event rate $\lambda_0$.

## Interpreting the generative model

The PIPPET generative model is formulated as though it implements perfect variational Bayesian inference on inherently stochastic stimuli. However, Bayesian computations in the brain are often invoked to compensate for internal as well as external sources of stochasticity [49], and in the case of PIPPET the most reasonable interpretation may be a combination of the two possibilities. In reality, we do not often listen to musical rhythms with random timing and phase jitter; however, neural noise and interaction with other ongoing processes may introduce timing variability into the processing of sensory events and give rise to variability in the process of tracking estimated phase. This interpretation also allows for changes in generative model parameters based on internal states that might affect internal noise levels, e.g., attentiveness (which has been shown to affect tempo correction but not phase correction [50], and which therefore might be modeled through its effect on $\sigma_\theta$). Ideally, the phase inference framework could be reconstructed based on assumptions of a combination of internal and external noise; however, that is beyond the scope of the current work.

Given this ambiguity, the generative model parameters may ultimately reflect some combination of the empirical statistics of rhythmic stimuli and internal factors. We briefly discuss the precision parameters $v_i$ as an example. First, an upper bound on the precision of expected event timing is the precision of sensory timing perception, which is, for example, high for human audition and significantly lower for human vision (An event can only be experienced after it occurs, so (as pointed out in [21]) the likelihood function on underlying phase associated with this type of uncertainty should be asymmetrical. The analytically tractable incarnation of our framework presented here uses Gaussian likelihood peaks, so cannot account for the effect of asymmetrical likelihoods; however, we could posit a $\lambda$ function with asymmetrical peaks and use numerical methods rather than the explicit solution derived here to estimate underlying phase at each time step). Second, expected event timing precision may further reflect the observed relative timing distributions of event streams. These observations may inform expectations on time scales ranging from a single sitting to a lifetime of listening. Expected timing may be learned separately for different sensory modalities, different musical genres (e.g., techno vs. funk), or even different instruments (e.g., kick drum, snare, hi-hat, as discussed below). The precision of a beat-based temporal expectation is closely related to the width of a "beat bin," the window of time (rather than a single time point) that is proposed to

constitute the "beat" in [51], and to the width of the temporal "expectancy region" described in dynamic attending theory [11]; in both cases, this width is increased by imprecision in the immediately preceding stimulus.

## Testable behavioral predictions

Given the ambiguous interpretation of the generative model discussed above, the question of whether human expectation-based entrainment is truly described by a normative framework may be ill-posed. However, two key qualitative elements of this framework can be tested directly: the tracking of phase uncertainty and the influence of expectations in the absence of events. Seeking further experimental evidence of these two phenomena would help determine the value of phase-inference-based models in describing human entrainment behavior.

The phase inference framework predicts that the accumulation of uncertainty over the course of empty time has a critical effect on the perceptual interpretation of subsequent events. In Fig 4, we show a rhythm that is perceptually misinterpreted due in part to empty time preceding syncopation. An experiment could be designed along the lines of [33] to test this aspect of the phase inference framework by measuring the effect of empty time on the interpretation of rhythmic stimuli that follow.

A second prediction along these lines is that various measurable perceptual phenomena, including period-dependent error correction in motor entrainment, perceptual parsing of ambiguous rhythms, and susceptibility to temporal illusions such as the filled duration illusion, should depend critically on levels of phase and tempo uncertainty. Assuming that the parameters of uncertainty tracking vary across individuals, the PIPPET/PATIPPET framework would predict correlations in measurements across these domains: certain individuals should show increased sensitivity to temporal illusion, misleading rhythms, and the effect of period on error correction. Further, stimulus manipulations that affect phase and tempo uncertainty, including the temporal precision of the auditory events and the length of the click train establishing an initial tempo estimate, should have direct and predictable effects on these perceptual and behavioral measures.

Third, the phase inference framework predicts that omissions of strongly expected events should systematically distort estimates of phase and tempo, or, perhaps indistinguishably, of elapsed time. These effects could be explored by parametrically manipulating event expectations through priming stimuli and then measuring distortions induced by event omissions through perceptual report or timed motor response.

If we find situations in which human behavior qualitatively differs from solutions to the inference problems posed by PIPPET and PATIPPET, these can be interpreted in two perfectly valid ways: either human behavior has not been optimally tuned for the task at hand, or we have not correctly identified and encapsulated the task and its survival-relevant objective. If we follow the latter interpretation, we might attempt to refine the generative model, e.g., by introducing the belief that tempo changes occur in jumps or ramps rather than as random drift, or to modify the objective of the task, e.g., by including additional cost functions or priors associated with perceptual report or motor output as discussed above.

## Application to analysis of behavioral data

The phase inference framework offers a predictive processing lens for understanding the results of rhythm perception and production experiments. Given a perceptual or behavioral task, we can suppose that motor or perceptual human entrainment behavior is optimally solving an inference problem, and determine the parameters of that problem by fitting them with appropriate methods. These parameters come with natural interpretations in the language of

prediction and precision. We can then study the changes in these parameters over the course of an experiment, over different variations on the same experiment, over the human lifespan, across cultures, etc.

For some experimental data, the many parameters available in PIPPET may prove redundant. For example, the observation of weak error correction in entrained tapping could be explained by imprecise auditory timing expectations (high $v_i$), an overly precise internal model of phase (low $V_t$, caused perhaps by low $\sigma$), or overly precise tap feedback timing expectations (as discussed below). However, we believe these to be meaningful distinctions that call for disambiguation through carefully designed experiments—for example, skipping taps to separate out the precision effects of tapping feedback or varying silent durations within the stimulus to separate the accumulating effects of phase uncertainty $V_t$ from the history-independent effects of timing expectation uncertainty $v_i$. For experiments that do not take such measures, redundant parameter sets that fit the data may be interpreted as meaningfully different possible interpretations of the results.

**Multiple event characteristics.**   mPIPPET generalizes the PIPPET/PATIPPET framework to cases of multiple distinguishable event types, each with its own set of expectations as a function of phase. One example could be listening, tapping, or dancing to a kit drum track with bass drum, snare, and hi-hat cymbal. Timing perturbations of different instruments in drum rhythms have been shown to differently affect human entrainment [52]. By letting $j$ take values from {$bass,snare,hihat$} and choosing appropriate values for $\phi_i^j$, $v_i^j$, and $\lambda_i^j$ for each event $i$ on the metrical grid, one could create a set of timing expectations with strength and precision dependent on the specific drum and metrical position that could then be used to optimally track underlying phase and tempo through a complex kit drum rhythm. A similar setup could be used to implement the assumption that pitches in a melody match the harmonic context more often in strong metrical positions, allowing rhythm parsing during melody listening to be influenced by scale degree.

Alternatively, the $j$ index may be used to treat events over multiple sensory modalities. Visual event timing is judged with less precision than auditory event timing in perceptual report [21] and in timing-sensitive sensory pathways [53], and might therefore be modeled with a less precise expectation template. (Note, however, that visual information may not have the same access to motor-related brain regions used for auditory entrainment [54], so the same modeling framework may not be appropriate.)

mPIPPET with $j \to \infty$ can be used to account for a continuum of event types. Thus, we could create a forward model in which it is more likely for notes played with stronger accents to fall on strong beats, or in which lower pitches are expected with higher timing precision [55] and therefore exert greater influence on neural entrainment [56].

The phase inference framework could be further generalized to take into consideration additional stream of continuous input. This could be visual input from watching a pendulum, auditory input from a continuously modulated sound, or proprioceptive feedback from continuous entrained motion (as opposed to discrete, timed proprioceptive feedback like tapping). This goes beyond the scope of the mathematics presented here, but is a straightforward application of results proven in [23].

**Tapping.**   As illustrated in Fig 4, mPIPPET can be used to describe entrained tapping data. Experiments have shown that the presence of entrained tapping prior to temporal perturbations in a metronomic stimulus reduces the phase correction response [57], indicating that the estimate of moment-by-moment phase is influenced by the proprioceptive, tactile, and auditory feedback from tapping. The phase inference framework is well-suited to modeling this influence as its own separate stream of informative input, though a thorough tapping

model would require introducing noise into tap execution and into the phase tracking process itself.

Importantly, using tap times to inform an estimate of underlying phase challenges our interpretation of this phase representing a purely external source of temporally patterned events. Instead, the inferred phase would be a hybrid of an external phase and the phase of one's own motor cycle. Functionally, this is similar to the perceptual oscillator forced by both an external stimulus and one's own periodic action proposed by [58]. This may be an especially useful way to think about synchronization with another agent, where one can adopt strategies ranging from following (assigning high precision to input from the other) to leading (assigning low precision to input from the other, and possibly higher precision to self-generated events). See [59] for a discussion of such a coding strategy as a means of minimizing representational neural resources.

**Aperiodic rhythm, speech, and musical grammar.**   One specific question that the phase inference framework might help resolve is how periodic and nonperiodic entrainment differ. PIPPET does not intrinsically differentiate between these two processes; however, since it is sufficiently general to model both, it could guide an exploration of parameter differences between the performance of similar tasks in periodic and aperiodic contexts. (For neural and behavioral evidence of differences between memory-based and periodicity based entrainment, see, e.g., [6, 46].)

By accommodating aperiodic expectations with any degree of precision or imprecision, the phase inference framework may be especially well-suited to modeling the loose temporal regularities of speech [60]. However, as currently formulated, it is limited in that expectations are not history-dependent: the occurrence or absence of an event does nothing to the expectancy of an event at a later timepoint. This is appropriate for modeling the metrical aspect of rhythmic expectancy, but does not address the grammar-like structure of music rhythm [61], i.e., the expectation of certain temporal patterns of events over others regardless of their metrical positions. Speech, of course, is even more thoroughly grammatical, with certain sound events strongly shaping the temporal and spectral patterns expected in the immediate future.

Such effects could be readily incorporated into the phase inference framework by adding history dependence to the expectation template $\lambda$, though that is beyond the scope of this work. The precise details of this history dependence in rhythm parsing could be based on any suitable formal model for rhythmic grammar (e.g., [61–63]), and for speech applications could include whatever aspects of the co-dependence of timing and content expectations were appropriate for the task at hand.

## Limitations and possible extensions of the phase inference framework

**Perceptual vs. motor entrainment.**   PIPPET is formulated as a perceptual process, without specific reference to how entrained movement is produced by this process. In presenting the PIPPET framework and using it to explain tapping results, we have posited that perceptual and motor entrainment are rooted in the same internal tracking of the phase of an external process. However, perceptual and motor measures of entrainment sometimes give conflicting results: for example, exposure to musical performance with expressively irregular timing affects perceptual reports of timing in subsequent stimuli [64], but does not affect phase correction in tapping to subsequent stimuli [65].

We expect that both physical entrainment and perceptual report are informed by a neural process of estimating underlying phase. Principles of economy suggest that they should share in such an estimate rather than drawing on separately instantiated processes of neural inference, and experimental correlations between motor and perceptual results tentatively support

this conclusion (e.g., [66]). However, it is possible that rapid, automatic audiomotor adjustment mechanisms have been selected to prioritize speed over precision (e.g., the spinocerebellar vermis [67]), especially in the case of entrainment to simple isochronous stimuli, and thus may not take uncertainty into account. If this is the case, then motor entrainment experiments not be clean indicators of perceptual management of uncertainty until the effects of these mechanisms are separated out.

**Learning expectation templates.** If the brain does treat entrainment as a process of inference based on a generative model, this raises the question of how the properties of the generative model are established in the first place. The PIPPET framework does not address this question directly, but by examining the parameters necessary to formulate PIPPET, we can clearly see what components need to be in place before a process of continuous phase and tempo updating can begin.

First, the brain must learn the temporal structures of the expectation template for rhythmic expectation. Learning these underlying structures from an experiential corpus of noisy, complex rhythms is not trivial. It seems likely to involve some type of bootstrapping in which a recognition of some degree of temporal structure allows for attribution of events to positions in that structure, allowing for deeper structure learning. Earlier exposure to simpler, less complex rhythms would likely help with such a bootstrapping process. (For a discussion of the challenges of this type of simultaneous learning and filtering and a proposed solution for non-point-process data, see [68].)

The brain must also learn noise and precision parameters for the model. Note that neither the temporal expectation variance parameters $v_i$ nor the noise parameter $\sigma$ necessarily correspond to the actual precision of the neural or external timing mechanisms in play. The brain may underestimate the noisiness of the timing process it uses to track underlying phase, leading to under-adjustment to auditory event timing and minimal time-warping between events, or do the opposite. Presumably, these parameters must be learned through experience and prediction error.

**Selecting and updating expectation templates.** When the brain is exposed to a rhythmic stimulus, it must first recognize that a predictable pattern exists and select an appropriate expectation template from its learned repertoire. This is its own process of inference, and may be amenable to a Bayesian description [69]. Since the PIPPET filter maintains a unimodal posterior, it is not well-suited to model this initial inference process, which may require maintaining a distribution over multiple distinct possible starting phases and expectation templates. This problem might be partially addressed by incorporating a model that evaluates multiple distinct hypotheses for beat or meter (e.g. [70, 71], or [72] with appropriate probabilistic interpretation) as an additional level of inference in parallel with ongoing phase and tempo inference.

## PIPPET in the brain

Though PIPPET and PATIPPET are abstract models not committed to a particular brain-based implementation, advances in the brain basis of timing and beat-keeping combined with the hypothesized neural bases of predictive processing suggest the beginnings of a plausible approximation of PIPPET in the brain, described below.

The essential aspect of the PIPPET framework that qualitatively differentiates its behavior from previous models is the explicit tracking of uncertainty over time for the purpose of informing the relative weights of sensory event timing and internal state estimates. There have been various proposals of how uncertainty is represented and utilized in the brain, and the system likely differs by task and type of uncertainty [49, 73]. One proposal is of particular interest

in relation to timing: uncertainty about a hidden state may be computed in medial frontal cortex and signalled via dopaminergic neurons in the ventral tegmental area [74]. In this case, the hidden state would be the phase and tempo of the stimulus. This proposal is consistent with the observations that dopaminergic neurons encode of certainty in the temporal expectation of sensory cues [75] and that dopamine receptor antagonism in humans causes increased timing uncertainty [76].

In the predictive processing literature, dopamine is often given the role of signaling certainty ("expected precision") across levels of hierarchical processing [77]. In this framework, it participates in probabilistic computations by weighting the input to error-calculating neural populations, causing these errors to be weighted more heavily in the ongoing process of error-minimization that implements variational Bayesian estimation of hidden states. Different dopaminergic populations may signal precision at different levels of processing; in particular, dopamine may signal precision of both higher-level state estimates and lower-level sensory expectations. Thus, phase certainty $V_t^{-1}$ and expected timing precision $v_i^{-1}$ may both influence computation through dopaminergic signalling.

Experiments with non-human primates have shown neural trajectories in medial premotor cortex (MPC, encompassing the supplementary and pre-supplementary motor areas) that represent progress through self-generated behavioral processes. The author hypothesizes in [78] that similar trajectories represent rhythmic phase in human MPC. A representation of a linear phase $\phi$, used in the phase inference framework for flexibility and mathematical tractability, would seem to be a limiting factor for implementation in the brain. For shorter, aperiodic learned patterns of temporal expectation, phase could be represented by short, aperiodic trajectories [79], as observed in primates in timed response tasks; for simple periodic patterns, phase could be represented circularly [80], as observed in isochronous tapping tasks; and for longer, hierarchical patterns, phase could be represented by hierarchically structured trajectories that loop but also evolve in other dimensions, as observed in cyclic behaviors whose sensory components change from one cycle to the next [81].

Guided by the "Action Simulation for Auditory Prediction" (ASAP) hypothesis presented in [82] and further developed in [78], the theory of hierarchical predictive processing [83], and the predictive functions proposed for the dorsal auditory pathway [84, 85], we propose a neural implementation of PIPPET's phase estimation in Fig 8. An essential aspect of this account is that it does not insist on the mathematical convenience of instantaneous phase updates, which are obviously implausible in the brain. Instead, precise timing predictions are issued with appropriate timing to intercept rising sensory signals, and the resulting timing errors are then be used to update phase through an error minimization process over the next few hundred milliseconds.

Briefly, phase is represented by stereotyped trajectories of population firing rates in MPC, and phase uncertainty is also represented locally in medial frontal cortex [74]. Basal ganglia selects and activates an expectation template appropriate to the context. This template is combined with phase and phase uncertainty estimates in MPC to compute a momentary subjective hazard rate $\Lambda$. The hazard rate is sent to parietal cortex as a prediction of event-based input, where it meets ascending pulses from the auditory system associated with auditory events (which may be relayed rapidly from the dorsal cochlear nucleus via cerebellum [19]). "Event prediction error" from parietal cortex returns to MPC, where it pushes $\mu$ in the direction that reduces error: toward expected event phases at events and away from them between events. This influence is opposed by a "phase prediction error" signal within MPC that pulls $\mu$ to progress steadily at tempo $\theta$. This error signal is weighted by phase precision $V^{-1}$.

Note that this is not a full, formal error-minimization scheme for implementing PIPPET, which is beyond the scope of this manuscript. In particular, it leaves out an updating scheme

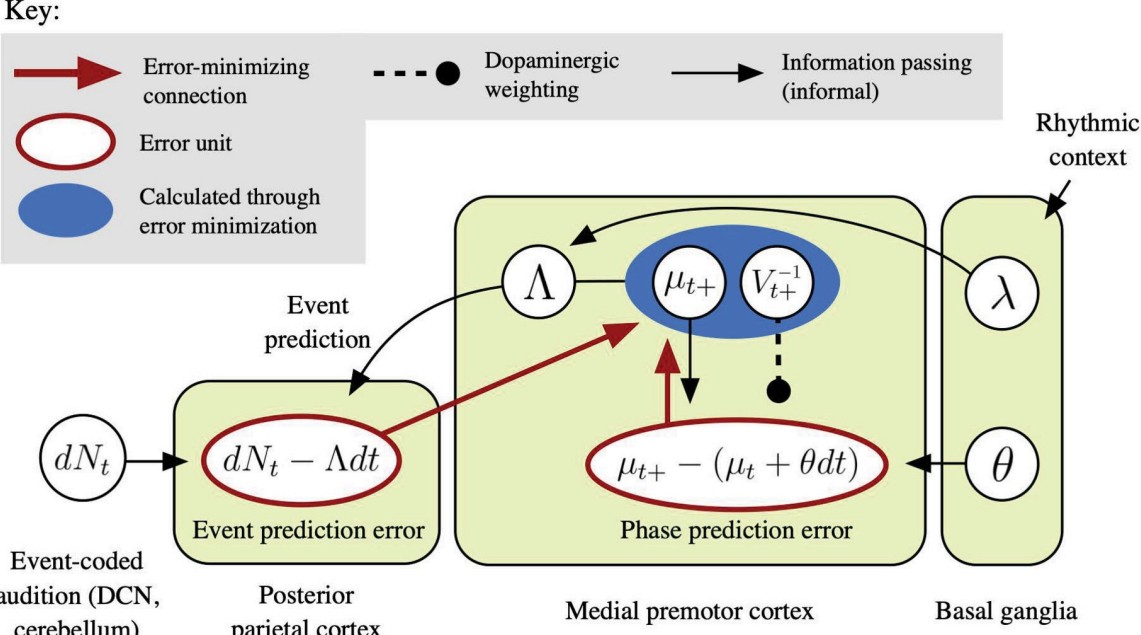

**Fig 8. A possible implementation of PIPPET in the brain.** This diagram embeds a formal predictive coding error minimization scheme is embedded within an informal information-passing schematic to outline how estimated phase $\mu_t$ might be calculated and updated on each $dt$ time step by a network of interacting brain regions. Estimated phase $\mu_t$ and phase uncertainty $V_t$ are represented in medial premotor cortex (MPC). These estimates are used to calculate instantaneous subjective hazard rate $\Lambda$ with the help of basal ganglia, which has selected an expectation template $\lambda$ based on recent rhythmic context. The hazard rate is sent to parietal cortex, where it acts as a prediction of pulses rising from the event-based auditory pathway. An "event prediction error" signal comparing pulses to their prediction is sent back up to MPC, where it pushes $\mu_{t+}$ in the direction that reduces prediction error—strongly toward local expectancy peaks when events occur, and weakly away from them when there are no events. (Note that phase updating at events is assumed to be rapid but not instantaneous as represented in the PIPPET filter.) The event prediction error is counterbalanced by a local "phase prediction error" signal generated through local interactions within MPC that pushes $\mu_{t+}$ to continue its steady forward progress. Phase prediction error is weighted by dopaminergic signaling of state precision $V_t^{-1}$ through VTA.

for $V$; see [86] for a discussion of the neurophysiology of precision updating. Further, it does not yet include an appropriate scheme for weighting event prediction error.

Although it would be difficult to directly test this neurophysiological setting of PIPPET in humans, it may be possible to indirectly observe a PIPPET-like process in neural data. At the scalp level and in intracortical electrodes, slow electrical oscillations do seem to anticipatorily track the structure of periodic auditory stimuli [87, 88], and this tracking is associated with the subjective passage of time [89]; these oscillations could be explored as possible estimates of mean underlying phase, with particular focus on those in motor areas. Ideally, timing prediction errors could be observed in the evoked EEG response to events (the ERP), allowing a direct measurement of event expectancy at each event time, and there are indeed indicators that the ERP is sensitive to temporal predictability (e.g., [90, 91]); however, the sensitivity of the ERP to recent stimulus history makes this approach unpromising. However, timing prediction errors may be observable in EEG/MEG through their effect on gamma oscillations [92, 93]. Further, the subjective hazard rate $\Lambda$ itself may be observable by using techniques recently applied to decode the temporal hazard function from EEG data [94], or through its correlation with beta oscillations [95].

Although human-like beat-based perceptual and audio-motor entrainment seems to be unique to humans, other primates do show rudimentary rhythmic timing abilities, especially in the visual modality [96], and represent phase of self-generated cyclic behavioral processes in

MPC [80, 81]. Experimental paradigms appropriately modified to engage mechanisms of self-action tracking might activate in non-human primates the same mechanisms of uncertainty-informed event-timing-based phase tracking that we hypothesize for auditory rhythm tracking in humans. Thus, primate neurophysiology in MPC and the dopaminergic system may be a promising avenue for indirectly testing the phase inference framework as a description of the human faculty of rhythmic entrainment.

## Supporting information

**S1 Text. The PATIPPET filter.** Equations describing the filter used to infer both phase and tempo based on point process event timing and a set of event timing expectations over phase. (PDF)

**S2 Text. Derivation of filter equations.** Mathematical process for deriving the PIPPET and PATIPPET filters from existing point process filtering equations. (PDF)

**S3 Text. Simulation parameters.** All parameters used in simulation to create the figures above. (PDF)

## Acknowledgments

Thanks to Tom Kaplan for extensive discussions and key insights motivating this manuscript, and to Aniruddh Patel, Darren Rhodes, and Nori Jacoby for helpful feedback.

## Author Contributions

**Conceptualization:** Jonathan Cannon.

**Formal analysis:** Jonathan Cannon.

**Investigation:** Jonathan Cannon.

**Methodology:** Jonathan Cannon.

**Project administration:** Jonathan Cannon.

**Visualization:** Jonathan Cannon.

**Writing – original draft:** Jonathan Cannon.

**Writing – review & editing:** Jonathan Cannon.

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
