## [Decision Letter · Decision Letter 0]

18 Jan 2021

Dear Dr. Cannon,

Thank you very much for submitting your manuscript "PIPPET: A Bayesian framework for generalized entrainment to stochastic rhythms" for consideration at PLOS Computational Biology.

As with all papers reviewed by the journal, your manuscript was reviewed by members of the editorial board and by several independent reviewers. In light of the reviews (below this email), we would like to invite the resubmission of a significantly-revised version that takes into account the reviewers' comments.

The reviewers have raised some very important issues that must be thoroughly addressed to render this paper suitable for PLoS Computational Biology.  If they are beyond the scope of this paper, then this manuscript may be more suitable for a journal oriented to more purely theoretical work. But if they can be addressed, then we will be happy to consider the revised MS for PLoS CB.  Specifically, the reviewers appropriately ask that a neurally mechanistic implementation for the model should be presented, that the biological insights gained from the work should be clearly stated, that additional parametric analysis of the model and how it functions should be provided, and that experiments that could be used to falsify the model should be proposed.  Of course, the reviewers' additional comments should also be addressed in revision.

We cannot make any decision about publication until we have seen the revised manuscript and your response to the reviewers' comments. Your revised manuscript is also likely to be sent to reviewers for further evaluation.

Sincerely,

Jonathan Rubin

Associate Editor

PLOS Computational Biology

Samuel Gershman

Deputy Editor

PLOS Computational Biology

Reviewer's Responses to Questions

**Comments to the Authors:**

Reviewer #1: In the current work the author introduces a Bayesian framework describing the human ability to entrain to an auditory stimulus. The work is clearly written, the model is well formulated and the author shows that the proposed framework qualitatively reproduces a set of empirical observations. However, the current study does not reach the required biological or methodological insight to be published in PLOS Comp. Biol. While it is a well sound work it belongs to a more mathematically (or modelling) oriented Journal.

While the proposed framework is compatible with some observations it is not clear how it can be mechanistically implemented at brain level neither can the author propose a feasible experiment capable of falsifying the model. Specifically, some abstractions of the model reducing its biological relevance are: the parameters are arbitrary adjusted with no clear biophysical interpretation, the predictions are not quantitatively compared with experimental data, there is no commitment for a neural mechanism, neither for a brain area, responsible for the Bayesian estimation, the same framework informs production and perception neglecting the interaction between the systems. Given that the framework is disconnected from the underlying biological process it is not clear for this Reviewer which are the new insights of this study on how the brain mechanistically works.

Author states in the Introduction that oscillator like theories describe predictions about the "what" rather than the "when". There are, however, examples in the literature where oscillator like models are used to make temporal predictions (see for example Assaneo, Rimmele et al. 2020 Nat. Hum. Beh.)

Reviewer #2: the review is uploaded as an attachment

Reviewer #3: This study proposes three models, PIPPET, PATIPPET and m-PIPPET using random diffusion variables to continuously estimate the underlying phase and tempo of beats with simple and complex metrics based on precise event times and their correspondence with timing expectations. These models estimate the posterior probability of phase and tempo using a likelihood distribution that represents a temporal expectation template and a prior function that represents the phase/tempo before to estimation. For PIPPET, the likelihood distribution is the sum of a linear combination of Gaussian functions centered on training phases, and a bias term. This function allows to consider different types of expected rhythmic events, such as isochronous or complex beats by adjusting the coefficients of the likelihood function. In addition, the a priori distribution is a Gaussian distribution with mean and variance equal to mean and variance of posterior distribution of the previous phase, this maintains a memory process over time.

Interestingly, the Snyder's theory is used to estimate the mean and variance of the posterior distribution in a continuously manner using the likelihood and the prior. The posterior distribution changes with the occurrence of an event as follows: 1) If an event is very close to a high expected phase, then the variance is reduced, emulating a restart of time processing. 2) if the event occurs in an expected phase neighborhood, the mean distribution is readjusted to the expected phase, emulating a time step readjustment. And 3) if an event occurs far from any expected phase, it has no effect on posterior distribution, this is convenient because in the absence of an event, the phase variable follows the course of time. These three elements reply synchronous or asynchronous rhythms and expected templates and describes very well time distortions.

For the PATIPPET model, the estimate of the expected interval duration is added. A re-adjustment in duration is observed minimizing error and temporal uncertainty. These models reproduce period-dependent phase corrections, illusory contraction of unexpectedly empty intervals, and failure to track excessively syncopated rhythms. In conclusion, this work shows an innovative model for continuously estimating the passage of time and duration of the interval between rhythmic events.

This is an interesting and well written and properly structured manuscript providing a novel set of models to continuously estimate performance on a wide range of timing and entrainment tasks, incorporating active inference on rhythmic timing. My main concerns on the paper are the following. First, there is no parametric analysis of the model variables and their impact on the solution of the posterior probabilities. In addition, a graphical representation of the intuitions described in page 10 would be very useful. Second, although the diffusion process has been well accepted biologically, with ramping activity of neurons as a plausible neural correlate, it is not clear how the brain can compute the mean and variance of the likelihood function, what is the neural correlate of the prior and how the posterior distribution can be computed from these two independent signals. Is not clear neither whether these computations depend on changes in discharge rate of a network, on the state dynamics of neural populations, and/or on the changes in power of oscillatory local oscillatory activity (see Balasubramaniam et al., 2020 J Neurosci). The author claims that the proposed models are more physiologically plausible than previous ones; however, this is not evident. In fact, the actual hypothesis is that rhythmic perception and entrainment depends on the close interaction of the audiomotor system, with a predictive signal generated in the motor system and a sensory input in the auditory cortex subject of active inference and error correction. The model integrates prediction, inference and error correction in the same system with no compartmentalization in the processing. In addition, the author simulates experiments of beat perception and beat entrainment interchangeably with the models, when we know there is clear differences in the neural processing (Merchant et al., 2015 Philos Trans R Soc Lond B Biol Sci). Furthermore, the time lag of the prior is only -1, what happens when you have different and multiple lags. Finally, the paper does not include a section on how the model parameters can be adjusted to experimental data, which can give a clear intuition on how the models are built.

Minor comments.

Please change the color for ∑t- and ∑t+ for Figure 2B.

A new panel in Figure 2 with ∑ and log(��) as a function of t could be very use full.

Figures 3, 4 and 5 should have a panel ∑ as a function of rhythm and expectation.

The legend of figure 3 instead of figure 5 should describe the specifics of the panels content.

**Have all data underlying the figures and results presented in the manuscript been provided?**

Reviewer #1: None

Reviewer #2: None

Reviewer #3: **No: **

PLOS authors have the option to publish the peer review history of their article (what does this mean?). If published, this will include your full peer review and any attached files.

Reviewer #1: No

Reviewer #2: No

Reviewer #3: **Yes: **Hugo Merchant
---

## [Decision Letter · Decision Letter 1]

29 Apr 2021

Hi Jonathan -

I am pleased to inform you that your thorough and careful revisions were a big hit with the reviewers.  You can see the official message below.

best,

Jon

Dear Dr. Cannon,

We are pleased to inform you that your manuscript 'Expectancy-based rhythmic entrainment as continuous Bayesian inference' has been provisionally accepted for publication in PLOS Computational Biology.

Best regards,

Jonathan Rubin

Associate Editor

PLOS Computational Biology

Samuel Gershman

Deputy Editor

PLOS Computational Biology

Reviewer's Responses to Questions

**Comments to the Authors:**

Reviewer #1: The author solved all my concerns.

Reviewer #2: The author has properly addressed my previous comments.

Reviewer #3: The paper has been properly modified to include all the reviewers’ comments. The paper now not only explains the structure of the models and their capabilities to track the beat under a wide variety of input circumstances, but also offers a set of predictions that can be tested in humans and a potential neural substrate for all the intervening key parameters computed on PIPPET. Importantly, the author included in the new version of the manuscript a graphical representation of the model in Figure 1, which gives a nice intuition about the key players of PIPPET. In addition, the author included a parametric analysis of the model variables and their influence on the calculation of the posterior probability estimators (Figure 2). It shows the systematic change of mean μ_(t+) and variance V_(t+) of the posterior probability in terms of expected framework variables (variance v_1 and influence λ_1 of the expected event) and prior distribution variables (mean μ_t and variance of V_t). It is clear now how the values of the model variables influence the calculation of the event timing estimates.

Minor comments.

Please clarify this sentence (line 125) “Here, we use event timing to inform a continuous variational inference process by first creating a generative model describing the probabilistic generation of precisely timed events and then variationally inverting that model. To model event generation, we use the mathematical tool of point processes.

Typos

Line 242 Figure Figure 2, column i

**Have the authors made all data and (if applicable) computational code underlying the findings in their manuscript fully available?**

Reviewer #1: Yes

Reviewer #3: Yes

PLOS authors have the option to publish the peer review history of their article (what does this mean?). If published, this will include your full peer review and any attached files.

Reviewer #1: No

Reviewer #2: No

Reviewer #3: No

**Have all data underlying the figures and results presented in the manuscript been provided?**

Reviewer #2: None

---

## [Editor Report · Acceptance letter]

25 May 2021

PCOMPBIOL-D-20-02007R1 

Expectancy-based rhythmic entrainment as continuous Bayesian inference

Dear Dr Cannon,

I am pleased to inform you that your manuscript has been formally accepted for publication in PLOS Computational Biology. Your manuscript is now with our production department and you will be notified of the publication date in due course.

With kind regards,

Katalin Szabo
